# Candidacy for Cochlear Implantation in Prelingual Profoundly Deaf Adult Patients

**DOI:** 10.3390/jcm11071874

**Published:** 2022-03-28

**Authors:** Ghizlene Lahlou, Hannah Daoudi, Evelyne Ferrary, Huan Jia, Marion De Bergh, Yann Nguyen, Olivier Sterkers, Isabelle Mosnier

**Affiliations:** 1Unité Fonctionnelle Implants Auditifs, Département d’Oto-Rhino-Laryngologie, Groupe Hospitalo-Universitaire Pitié-Salpêtrière, APHP Sorbonne Université, 75013 Paris, France; hannahpaule.daoudi@aphp.fr (H.D.); evelyne.ferrary@inserm.fr (E.F.); marion.de-bergh@aphp.fr (M.D.B.); yann.nguyen@aphp.fr (Y.N.); o.sterkers@gmail.com (O.S.); isabelle.mosnier@aphp.fr (I.M.); 2Technologie et Thérapie Génique de la Surdité, Inserm/Institut Pasteur, Institut de l’Audition, 75012 Paris, France; 3Shanghai Key Laboratory of Translational Medicine on Ear and Nose Diseases (14DZ2260300), Department of Otolaryngology-HNS, Shanghai Ninth People’s Hospital, Shanghai Jiao-Tong University School of Medicine, Shanghai 200025, China; huan.jia.orl@shsmu.edu.cn

**Keywords:** cochlear implant, prelingual profound hearing loss, speech perception, quality of life

## Abstract

Cochlear implantation is usually not recommended for prelingual profoundly deaf adults, although some of these patients might benefit from it. This study aims to define the candidates for cochlear implantation in this population. This retrospective study reviewed 34 prelingual profoundly deaf patients who had received a cochlear implant at 32 ± 1.7 years old (16–55), with at least 1 year of follow-up. Speech perception and quality of life were assessed before and 3, 6, and 12 months after cochlear implantation, then every year thereafter. According to the word speech intelligibility in quiet (WSI) 1 year after implantation, two groups were identified: good performer (GP) with WSI ≥ 50% (*n* = 15), and poor performer (PP) with WSI ≤ 40% (*n* = 19). At the 1 year mark, mean WSI improved by 28 ± 4.6% (−20–100) (*p* < 0.0001). In GP, the intelligibility for words and sentences, communication and quality of life scales improved. In PP, the communication scale improved, but not auditory performance or quality of life. GP and PP differed pre-operatively in speech production, communication abilities, and WSI in best-aided conditions. In prelingual profoundly deaf adults, a dramatic auditory performance benefit could be expected after cochlear implantation if the patients have some degree of speech intelligibility in aided conditions and have developed oral communication and speech production.

## 1. Introduction

Cochlear implantation is a well-established treatment to restore communication in patients with post-lingual severe-to-profound hearing loss regarding their age, and in prelingually deaf infants [1,2,3]. Because of the short sensitive period for language development in infants, the shorter the auditory deprivation, the better the auditory performance and oral language acquisition that would be achieved after cochlear implantation [4,5]. Indeed, late cochlear implantation (after 5 years of age) is usually not recommended for non-progressive congenital or prelingual profound hearing loss [6,7], as prelingual profoundly deaf patients implanted when adults achieved lower auditory performance with their cochlear implant than those of post-lingual cochlear implanted users [8,9], although several studies have emphasized some benefits in auditory scores [8,10,11,12,13,14,15], quality of life [16,17], or self-esteem [11] in these late-implanted prelingual profoundly deaf patients. However, some patients implanted when adult who achieved poor performance after cochlear implantation have been reported to abandon the device [13], raising the question of who would be the prelingual profoundly deaf adult candidates to achieve sufficient benefit for a daily use. In a recent systematic review of the literature, there were not enough data to predict the auditory performance and analyze the prognosis factors of cochlear implantation outcomes in this population [18,19]. Most of the studies were small and heterogenous, including late-implanted prelingually and/or perilingually deaf patients, presumably because of the few validated indications to date for cochlear implantation in these populations.

The aim of the present study is to analyze cochlear implantation outcomes in a population of prelingual profoundly deaf adults to define the putative candidates who would attain a real benefit in daily life.

## 2. Materials and Methods

### 2.1. Patients

A monocentric retrospective cohort study was conducted in a tertiary referral center and included patients with unilateral or bilateral cochlear implant for a prelingual bilateral profound hearing loss, which were implanted over 15 years old without an upper age limit, between 2004 and 2019. The decision for a cochlear implantation was made after a multidisciplinary evaluation that included clinical, audiometric, speech, psychological and radiological assessments, and in accordance with the national guidelines for cochlear implantation [6,7,20], i.e., a speech discrimination ≤50% at 60 dB for disyllabic words in silence in the best-aided conditions. Patients were selected among a cohort of all cochlear implant recipients operated on in our department and noted as “congenital hearing loss” (Figure 1). Selection criteria were: (1) prelingual hearing loss diagnosed before the age of 4 years old, reported as severe to profound at diagnosis according to the medical file; (2) postoperative follow-up of at least 1 year; and (3) preoperative audiometric data available. Unilateral or bilateral, either simultaneous or sequential, cochlear implantations were performed with 3 different cochlear implant brands from Cochlear™, Med-El™, and Oticon™. Decision making for a bilateral cochlear implantation was dependent on deafness history, patient’s request and motivation. All surgeries and devices were covered by government funds. Sequential bilateral cochlear implantation was performed with the objective of achieving a better performance, especially in noisy conditions.

All patients or parents of teenagers provided written informed consent allowing the analysis of their data in accordance with the reference methodology of the National Commission for Data Protection and Liberties (CNIL-France, N°2040854). The study complies with the Declaration of Helsinki code (“World Medical Association Declaration of Helsinki. Recommendations Guiding Physicians in Biomedical Research Involving Human Subjects”, 1997).

Data were collected on demographics, characteristics of hearing loss, pre-operative audiometric measurements, pre- and postoperative speech evaluation, and communication skills assessment. According to the medical charts, etiology, age and degree of hearing loss at diagnosis, age of first hearing aid fitting, and duration of hearing aid use were collected. A genetic cause of hearing loss was either proved by molecular analysis (Sanger), in the case of a bi-allelic mutation of *GJB2* (CONNEXIN 26 gene), or assumed to be a genetic-related hearing loss because of a familial history of similar hearing loss in a first-degree relative, or because of characteristic inner ear malformation (bilateral enlarged vestibular aqueduct). Socio-professional categories were classified into seven classes according to the main stages of French education (Poitrenaud’s scale—Appendix A) [21].

### 2.2. Auditory Performance and Communication Evaluations

Hearing was evaluated in a soundproof booth first with headphones and then in free field in quiet. Before implantation, pure tone average (PTA) and speech intelligibility without hearing aid using disyllabic words (Fournier lists) were measured with headphones. Speech intelligibility without and with lip-reading was evaluated in free field with a signal presented at 60 dB HL from the front, in best-aided conditions (with either uni- or bilateral hearing aids) before implant, and with cochlear implant alone after implantation. The following tests were systematically conducted in quiet: speech intelligibility using disyllabic words (Word Speech Intelligibility, WSI) and speech intelligibility for sentences (Marginal Benefit for Acoustic Amplification, MBAA). The results are expressed as the percentage of correct words for disyllabic lists, and as the percentage of correct words and sentences for the MBAA sentences.

The main outcome was WSI with cochlear implant alone 1 year after implantation. For simultaneous or sequential bilateral implantations, the evaluation was performed unilaterally on the best ear or first implanted ear, respectively. Patients were divided in two groups according to WSI with cochlear implant alone at 1 year: good performer (GP) when WSI ranged between 50% and 100%, and poor performer (PP) when WSI was less than 50%. A WSI below 50% was considered as a poor performance as it corresponds to the 25th percentile of all cochlear implant adult recipients in our center during the same period (*n* = 612, unpublished data).

Before implantation, auditory tests in noisy conditions were not performed because of the severity of hearing loss. In some instances, speech intelligibility achieved sufficient level 1 year after implantation to be tested in noisy conditions, which was performed with a speech-to-noise ratio of 10 dB (SNR10), and subsequently each year. For patients with a sequential bilateral implantation, a complementary analysis compared the auditory performance in best-aided conditions before and 1 year after the second implantation.

Communication skills were assessed before and after implantation using the Category of Auditory Performance scale (CAP, Appendix A) [22], scaled from 1 (no awareness of environmental sound) to 9 (can use the telephone with an unfamiliar talker). The usual mode of communication used before implantation was noted (code, sign language or oral). Moreover, the scores obtained with the APHAB (Abbreviated Profile of Hearing Aid Benefit) questionnaire were also studied, expressed from 0 (no difficulty) to 100% (maximal impact of hearing loss on communication) [23]. The language assessment was performed before implantation using the Speech Intelligibility Rating (SIR, Appendix A), scaled from 1 (pre-recognizable words in spoken language) to 5 (connected speech intelligible to all listeners; the patient is easily understood in everyday contexts) [24]. After implantation, the follow-up protocol was standardized, and results were assessed using speech intelligibility for disyllabic words and MBAA scores at 3, 6, 12, and 24 months, APHAB and CAP at 1 year, and every year thereafter.

### 2.3. Statistical Analysis

Statistical analysis was performed using GraphPad Prism^®^ (version 8.2.0, San Diego, CA, USA). Values were expressed as mean ± standard error of mean (SEM) (min–max). Two-way ANOVA, Wilcoxon, and Mann–Whitney tests were used to compare quantitative data, and Fisher’s and chi-squared tests were used for qualitative data. A *p* value < 0.05 was considered to be significant.

## 3. Results

### 3.1. Population

Thirty-four patients were included in the study (Table 1). Hearing loss was diagnosed at 21 ± 2.2 months (1–46), as severe and profound in 15 cases (44%) and 19 cases (56%), respectively. At the age of implantation (32 ± 1.7 years (16–55)), all patients were profoundly deaf (non-aided auditory thresholds >90 dB for both ears), and the mean preoperative WSI in best-aided conditions was 12 ± 2.4% (0–50). A confirmed or presumed genetic cause of hearing loss was found for 50% of patients (*n* = 17), including 12 patients with a confirmed mutation of *GJB2* (CONNEXINE 26 gene). The other etiologies were unknown for 11 (32%), intra-uterine infection for 4 (12%), a neonatal meningitis for 1 (3%), and a prematurity for 1 (3%). Implantation was unilateral for 18 patients (53%), bilateral sequential for 9 (26%), and bilateral simultaneous for 7 (21%). All preoperative data are detailed in Table 1.

### 3.2. Cochlear Implant Outcomes

Post-implantation follow-up was 4 ± 0.5 years (1–10). The mean WSI at 1 year was 36 ± 5.1% (0–100), and the median was 35% (interquartile range (7.5–60)). Compared to the preoperative scores, the mean WSI improved by 28 ± 4.6% (−20–100) (*p* < 0.0001, *n* = 34, Wilcoxon test). At 1 year post-implantation, 15 patients (44%) achieved a WSI of at least 50%, and the 19 others (56%) inferior to 50%; they were considered as good (GP) and poor (PP) performers, respectively (Table 1).

As shown in Figure 2A, WSI increased as early as 3 months post-implantation for GP to reach 65 ± 4.1% (50–100) at 1 year (*p* < 0.0001, two-way ANOVA, *n* = 15, Table 2). Excellent WSI scores ranging from 70 to 100% were achieved in seven cases, four being unilaterally implanted and the other three being contra-laterally operated on later (Table 1). For PP, no significant improvement was observed with a difference of +10 ± 3.3% (0–20) 1 year after implantation compared to preoperative scores (*p* = 0.052, two-way ANOVA, *n* = 19) (Table 2). Moreover, no word recognition was obtained in eight cases, half of them having been bilaterally and simultaneously implanted (Table 1). Whatever the auditory performance achieved after 1 year, WSI remained stable up to 10 years (Figure 2B).

Speech intelligibility for sentences without lipreading increased as early as 6 months post-implantation for GP achieving 64 ± 5.2% (33–100) at 1 year (*p* = 0.0007, two-way ANOVA, *n* = 12), but not for PP (Figure 2C, Table 2).

The addition of lip-reading did not change the WSI in the two groups: 74 ± 9.3% (0–100) (*n* = 14) vs. 98 ± 2.0% (90–100) (*n* = 5) for GP (*p* > 0.9, Wilcoxon test) and 58 ± 7.9% (0–90) (*n* = 17) vs. 56 ± 10.9% (0–100) (*n* = 13) for PP (*p* = 0.8, Wilcoxon test) before and 1 year after implantation, respectively.

Communication evaluated using the CAP score improved at 1 year post-implantation in both PP and GP (respectively *p* = 0.015 and *p* = 0.0003, chi-squared test, Table 2). At 1 year post-implantation, all patients of the GP group improved the CAP score, and 86% of them (*n* = 12/14 patients) reached a CAP ≥ 6, being able to understand common phrases without lip-reading. For PP, an improvement of the CAP score was achieved in 11/16 patients (69%), for whom CAP was available preoperatively and 1 year post-implantation, reaching a CAP ≥ 6 in only 35% of them.

Compared to the preoperative score, the APHAB score decreased 1 year after implantation for GP (−28 ± 3.6 (−38 to −18), *n* = 10, *p* < 0.0001), showing an improvement of their quality of life, and remained stable up to the second year of follow-up. No change on the APHAB score was observed for PP (Figure 2D).

Among the 34 implanted patients, 32 (94%) were all day long users, and time to time for the 2 others who were in the PP group. No patient has abandoned the device in the long term.

### 3.3. Analysis of Preoperative Factors

The 2 groups were similar in terms of age at diagnosis of hearing loss (*p* = 0.5, Mann–Whitney test), etiology of hearing loss (*p* = 0.2, chi-squared test), preoperative PTA (*p* = 0.3 and *p* = 0.08, respectively, for ipsi- and contralateral ear, Mann–Whitney test), age at first implantation (*p* = 0.7, Mann–Whitney test), and socio-professional categories (*p* = 0.2, chi-squared test) (Table 1). However, in best-aided conditions, preoperative speech intelligibility for disyllabic words without and with lip-reading was higher for GP compared to PP (*p* = 0.022 and *p* = 0.012, respectively, Mann–Whitney test, Figure 3A).

All patients except eight were equipped with a bilateral hearing aid at the time of diagnosis of deafness, with no significant difference between the two groups for age of the first hearing aid (*p* = 0.7 and *p* = 0.4, respectively, for ipsi- and contralateral ear, Mann–Whitney test, Table 1). Among the eight remaining patients (24%), six were PP: one patient did not use any hearing aid before cochlear implantation, two were equipped in only one ear at the time of diagnosis, and three were equipped several years after diagnosis. The last two were GP: one patient had a bilateral hearing aid at the age of 5 years, and one was equipped in only one ear at the time of diagnosis (Table 1). At the time of implantation, 29 patients (85%) used their hearing aid in the implanted ear up to surgery, except for PP, a patient who never used a hearing aid, and 3 who had abandoned their hearing aid, and for GP, 1 patient had abandoned the hearing aid 5 years before implantation. There was no difference in the duration of hearing loss without hearing aid between the two groups (*p* = 0.8, Mann–Whitney test, Table 1).

All patients, apart from 2 in PP, used oral communication before implantation, alone (20/34 patients, 59%) or in combination with sign language in (12/34 patients, 35%) (Table 1). More patients used only oral communication in GP compared to PP (87% vs. 37%, *p* = 0.005, Fisher’s test). Additionally, the preoperative SIR scores in GP were higher compared to PP (*p* = 0.003, chi-squared test, Figure 3B), 93% of GP patients having a connected speech intelligible to a listener who had no experience (score of 5) or little experience (score of 4) of a deaf person’s speech. Finally, the preoperative CAP scores in GP were high compared to PP (*p* = 0.03, chi-squared test, Figure 3C).

### 3.4. Benefits of Bilateralisation in the Case of Sequential Cochlear Implantation

A total of 9 patients (26%) had a sequential implantation, 5/15 and 4/19 in GP and PP, respectively, with a mean delay between the 2 implantations of 3 ± 0.9 (0.5–7.8) years. All these patients were still using their hearing aid in the non-implanted ear until the contralateral implantation. When comparing the auditory performance in best-aided conditions, no difference was found for WSI or for sentences intelligibility in quiet and in noise before and 1 year after the second cochlear implantation (Table 3). Additionally, there was no difference for APHAB score, nor for CAP score (Table 3). Although no difference was objectified, all sequential bilaterally implanted patients whatever the performance group used both cochlear implant processor all day long, and speech intelligibility scores in noisy conditions for sentences and words in sentences improved by more than 20% in GP (not significant), but not in PP (Table 3).

## 4. Discussion

In the present retrospective study, 34 adult patients with profound prelingual hearing loss were included and demonstrated a dramatic increase in auditory performance as evaluated 1 year after cochlear implantation with 1 cochlear implant alone using disyllabic words in quiet, regardless of whether patients were unilaterally or bilaterally implanted, although scores ranged from none to 100%. This is in line with other studies on series of patients demonstrating improvements for words [10,11,16], phonemes [10,13,25], and sentence recognition scores in quiet [10,11,12,13,16,17,26,27,28], although prelingual and peri-lingual hearing loss were in most cases mixed. This study reports among the largest cohorts of prelingually deaf adults cochlear implant recipients in the literature and analyzes the outcomes in terms of speech intelligibility for words and sentences, as well as in terms of quality of life and communication skills.

In this series, 44% of patients experienced a dramatic increase in auditory performance with a mean speech intelligibility in silence of 65 ± 4.1% (GP), similar to post-lingually implanted profoundly deaf adult scores (67% to 76% depending on the test used [1,29,30]). Accordingly, the speech intelligibility for disyllabic words in quiet of all the patients in the French cohort recently implanted at adult age reached 67% [30]. Having such good performance after a delayed cochlear implantation is a surprising fact, as it is commonly accepted that auditory performance after cochlear implantation in adults is significantly lower in patients with prelingual hearing loss compared to post-lingually acquired hearing loss, and that the duration of auditory deprivation is a major prognostic factor [8,9,31]. However, some prelingually deaf patients, who did not have a cochlear implant at an early stage, could really benefit from a cochlear implantation even if delayed, and obtain better speech intelligibility, as well as improved communication skills and quality of life.

For the other 56% of prelingually deaf patients with poor auditory performance (PP), a slight or no improvement was observed for disyllabic word intelligibility 1 year after the cochlear implantation with no further improvement, at variance with previously reported by others [32,33,34]. However, the everyday communication, as evaluated with the CAP score, was improved, and most of the recipients, except two, are all day long users. Moreover, some of them asked for a second cochlear implant on the other ear. Thus, judging the input of the cochlear implantation only on auditory performance does not give a complete picture of the individual benefit in this population.

Concerning subjective evaluation, the APHAB questionnaire, specific to hearing loss, quantifies the trouble experienced in communicating in everyday life situations, and is routinely used to evaluate post-lingual deaf adults. Prelingually deaf adults have a long history of coping with their hearing handicap and have developed communication strategies so that they are not as dramatically affected in their social communication as post-lingually deaf adults. In this study, they obtained better scores in the APHAB questionnaire compared to post-lingual hearing-impaired adults before implantation [30]. Therefore, this questionnaire is probably insufficient to highlight the benefit of cochlear implantation in the particular cases of prelingual poor performers. Other studies have shown that the main improvements for this deaf population were on primary sound processing, sense of safety, and self-confidence [11,16]. If the cochlear implant does not provide enough improvement for speech perception without lip-reading, it could be a real help in audiovisual conditions, processing environmental sounds, or at least as an alert device. The subjective benefit and preoperative expectations have to be taken into account to assess the real input of cochlear implantation in this population.

Based on this study, multiple preoperative factors could predict good auditory performance after cochlear implantation. First, preoperative speech perception in best-aided conditions, with or without lip-reading, appears to be a good preoperative prognostic factor, as already reported [10,15,28,35]. Second, preoperative communication, referring to the type of language used by the patient (code, sign language, or oral) and to the CAP score, appeared to be a major prognostic factor, as already mentioned in several studies [15,16,25,26,28]. In this study, most patients used oral communication, alone or in combination with signed language, showing probably some benefit of their hearing aid and a history of intensive speech therapy in childhood. Preoperative communication is probably an indicator of whether the auditory cortex has a quite normal organization or not, as it has been proved that visual processing causes a functional shift of the auditory cortex into the visual processing pathway [36]. In addition, the intelligibility of the patient’s speech appears to be a prognostic factor in this study as in others [15,16,25,35], suggesting that the better the patient’s speech was, the better their oral and auditory skills were developed in childhood. Third, hearing aid use until the day of cochlear implantation should also be taken into consideration before making the decision for cochlear implantation in an adult prelingually deaf patient, even though it does not appear to be a significant prognostic factor in this study, since the great majority of included patients wore their hearing aid until the cochlear implantation. However, most of the patients who did not wear hearing aids for a long period of several years did not achieve any word recognition. Finally, patient’s motivation has to be deeply analyzed before cochlear implantation. In this study, patients came with a request for improved communication, often in the context of a change in their everyday life, such as starting higher education, new work, family, of after meeting cochlear implant recipients. Based on these results, we propose a decision algorithm for cochlear implantation in adulthood in the case of prelingual deafness (Figure 4). A bilateral simultaneous implantation should be discussed with great caution even in front of an express demand of the patient and/or his/her family. Alternatively, the benefit of a bilateral sequential implantation has to be made on a longer period of follow-up than 1 year after the contralateral procedure to evidence better performance in noisy conditions than with a unilateral cochlear implant [37]. Whether a contralateral implantation should be restricted to good performers or to all the patients is still a matter of debate.

This study has some limitations, as it was a retrospective analysis, with some missing data, making the power of the study weaker. Additionally, a selection bias is possible since the diagnosis of hearing loss was made many years ago. In addition, some patients who were operated on several years earlier, especially patients using sign language alone, were lost to follow-up, or were not evaluated properly before implantation, and were not included in this study (Figure 1). Finally, it would be interesting to evaluate the duration of post-operative speech therapy reeducation, which is probably longer for patients with prelingual hearing loss compared to post-lingual ones.

## 5. Conclusions

Cochlear implantation could be considered an adequate option for adults with prelingual onset profound hearing loss who request it, after an exhaustive evaluation of speech intelligibility, communication, speech production and expectations of the patient. For patients with developed oral communication, good speech production, and some degree of speech intelligibility with their hearing aid, a dramatic benefit should be expected, making these patients good candidates for cochlear implantation at the adult age. For those who experienced no measurable benefit on speech perception, the use of the cochlear implant processor might improve their communication skills. A proper assessment, adapted to these non-conventional cochlear implantation candidates, is thus needed pre- and post-operatively for this particular population of prelingual profoundly deaf adults.

## Figures and Tables

**Figure 1 jcm-11-01874-f001:**
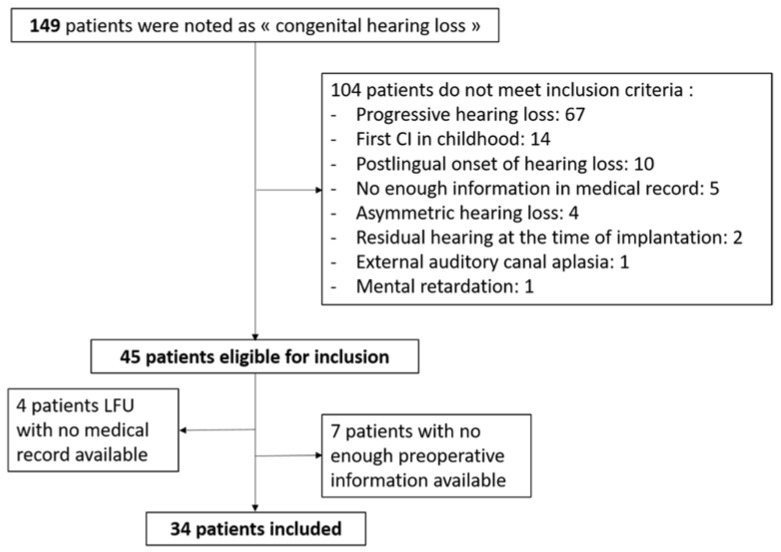
Flowchart of patient inclusion.

**Figure 2 jcm-11-01874-f002:**
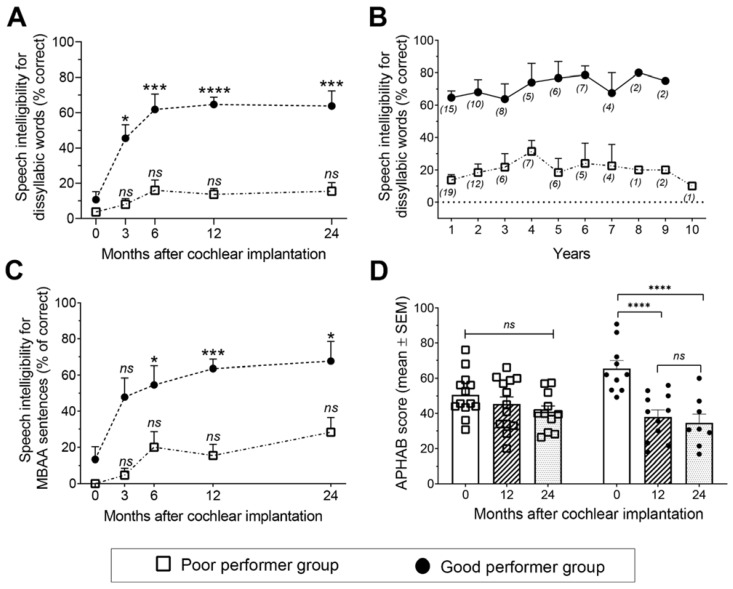
Post-operative evolution after cochlear implantation. (**A**) Speech intelligibility without lipreading for disyllabic words in silence in the 2 groups. Compared to the preoperative scores, an improvement was observed for the good performer group at 3 months (*p* = 0.015), 6 months (*p* = 0.0002), 1 year (*p* < 0.0001), and 2 years (*p* = 0.0005) post-implantation. For the poor performer group, no significant improvement was found. *n* = 15, 11, 11, 15, 8 for the good performer group, and 19, 10, 10, 19, 11 for the poor performer group, respectively at 0, 3, 6, 12, and 24 months after cochlear implantation. (**B**) Evolution of speech intelligibility for disyllabic words over time in the 2 groups. (*n*) are the number of patients at each time point. (**C**) Speech intelligibility without lipreading for MBAA sentences in the 2 groups. Compared to the preoperative scores, an improvement was observed for the good performer group at 6 months (*p* = 0.024), 1 year (*p* = 0.0007), and 2 years (*p* = 0.011) post-implantation. For the poor performer group, no significant improvement was found. *n* = 12, 11, 10, 14, 7 for the good performer group, and 9, 7, 8, 13, 10 for the poor performer group, respectively at 0, 3, 6, 12, and 24 months after cochlear implantation. (**D**) Mean ± SEM of APHAB scores before, 1 and 2 years after implantation in the 2 groups, showing an improvement only for the good performer group (*p* < 0.0001). APHAB, Abbreviated Profile of Hearing Aid Benefit. Data are mean ± SEM; a two-way ANOVA with a Tukey multiple comparisons test was performed for statistical analysis. * *p* < 0.05; *** *p* < 0.0005, **** *p* < 0.0001, *ns*: non significant.

**Figure 3 jcm-11-01874-f003:**
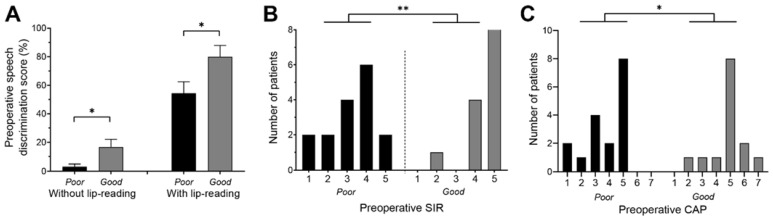
Preoperative assessments in the 2 groups. (**A**) Preoperative speech intelligibility for disyllabic words evaluated in optimal listening condition (i.e., with the 2 hearing aids if used) without and with lip-reading. Good performer group (GP) obtained better scores compared with to the poor performer group (PP) for the speech intelligibility without and with lip-reading (respectively *p* = 0.022, *n* = 15 for GP and 18 for PP, and *p* = 0.012, *n* = 13 for GP and 18 for PP, Mann–Whitney test). (**B**) Preoperative Speech Intelligibility Rating (SIR) scores. GP (*n* = 15) had better scores compared with PP (*n* = 16) (*p* = 0.003, chi-squared test, 3 missing values in PP). (**C**) Preoperative Category of Auditory Performance (CAP) scores. GP (*n* = 14) had better scores compared with PP (*n* = 17) (*p* = 0.032, chi-squared test, 1 missing for GP and 2 for PP). Data are mean ± SEM (**A**) and *n* (**B**,**C**). * *p* < 0.05; ** *p* < 0.005.

**Figure 4 jcm-11-01874-f004:**
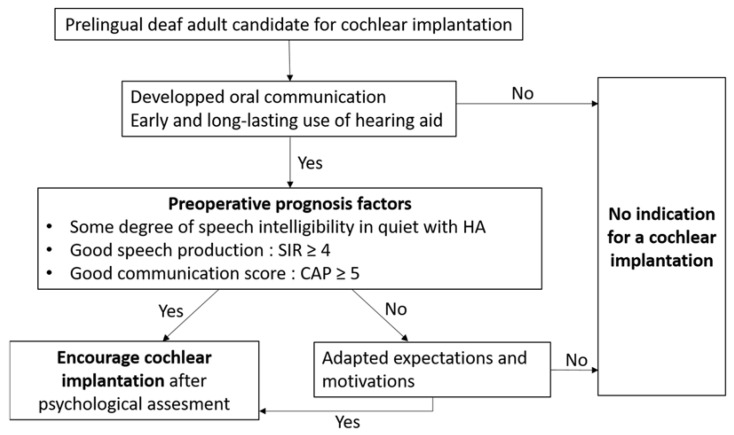
Decision algorithm for cochlear implantation in the case of an adult patient with prelingual severe-to-profound hearing loss. HA: hearing aid; SIR: Speech Intelligibility Response; CAP: Communication Auditory Performance.

**Table 1 jcm-11-01874-t001:** Characteristics of implanted patients classified in function of speech intelligibility of disyllabic word score in quiet (WSI) with one cochlear implant 1 year after implantation (best or first implant outcomes are given for simultaneous or sequential bilateral implantation, respectively). Good and poor performers are patients 1 to 15 and 16 to 34, respectively.

Patient	Sex	HL Etiology	HL Class at dg ^¥^	Age at dg (Months)	Age at First HA (Months)		Duration without HA (Years)	Social Category	CAP	SIR	Communication ^†^	Preoperative PTA (dB)		Type of Implantation	Age at CI (Years)	Year of First CI	CI Model	WSI at One Year (%)
					CI Ear	Non-CI Ear						CI Ear	Non-CI Ear					
1	F	Unknown	S	12	12	12	1	7	5	5	O	115	114	Unilateral	29	2019	CI522	100
2	F	Unknown	P	24	24	24	2	7	7	5	O	102.5	99	Bilateral seq.	22	2010	Ci512	90
3	M	Cnx 26	P	24	24	24	2	7	NR	5	O	120	111	Bilateral seq.	26	2015	CI24RE	80
4	F	Unknown	S	24	24	24	2	7	5	5	O	99	98	Unilateral	39	2014	CI24RE	70
5	F	Cnx 26	P	12	12	12	1	6	6	5	O	99	99	Unilateral	16	2013	CI24RE	70
6	M	Cnx 26	S	12	60	60	5	4	5	4	O	115	104	Unilateral	52	2018	CI522	70
7	F	IUI	P	1	3	3	0	7	4	5	O	115	102.5	Bilateral seq.	45	2013	Ci522	70
8	F	Unknown	S	24	24	24	2	6	5	5	O	109	99	Bilateral seq.	30	2012	Med-El Flex31	60
9	M	Meningitis	S	3	96	144	8	1	5	4	O	105	105	Bilateral sim.	47	2018	Neuro ZTI Evo	60
10	F	Cnx 26	P	16	16	16	1	7	6	5	O	102	96	Unilateral	32	2012	CI24RE	50
11	F	Genetic	S	40	40	40	3	7	5	4	O + S	109	112	Bilateral seq.	27	2012	CI422	50
12	F	IUI	P	9	9	9	5	2	2	2	O + S	117	111	Bilateral sim.	20	2007	Med-El	50
13	M	Cnx 26	S	15	16	16	1	7	5	4	O	111	109	Unilateral	32	2018	CI522	50
14	F	Unknown	P	24	24	24	2	7	5	5	O	109	99	Unilateral	27	2018	Ci522	50
15	M	Unknown	S	18	18	18	1.5	7	3	5	O	97.5	95	Unilateral	20	2013	Ci422	50
16	M	Unknown	S	9	9	9	1	7	3	5	O	111	99	Unilateral	31	2013	Ci422	40
17	F	Cnx 26	S	6	6	6	0	7	5	5	O	116	108	Bilateral sim.	39	2016	Ci512	40
18	F	Cnx 26	P	45	167	45	14	7	NR	NR	O + S	114	115	Unilateral	45	2004	Ci24CA	30
19	M	Unknown	P	18	18	18	1	6	5	4	O	99	91	Unilateral	21	2012	Ci24RE	30
20	F	Cnx 26	P	9	9	9	1	7	5	3	O + S	112	108	Bilateral seq.	25	2012	Ci24RE	30
21	M	Cnx 26	P	12	12	12	6	6	NR	NR	O + S	110	106	Unilateral	33	2007	Digisonic SP	20
22	M	Unknown	S	36	36	36	3	6	5	3	O + S	115	116	Unilateral	32	2008	CI24RE	20
23	F	Unknown	P	3	3	3	0	7	4	2	O + S	110	104	Unilateral	18	2011	CI24RE	20
24	F	Cnx 26	P	12	12	12	1	6	5	4	O + S	110	105	Bilateral seq.	30	2015	Ci422	10
25	M	Genetic	S	36	36	36	3	6	5	4	O + S	99	115	Unilateral	21	2014	Digisonic SP	10
26	F	Cnx 26	P	24	24	24	2	7	3	4	O + S	110	107	Unilateral	24	2016	Ci422	10
27	M	Genetic	S	46	46	46	4	6	5	5	O	115	110	Bilateral seq.	41	2010	Ci24RE	0
28	M	Premature	S	24	29	29	2	3	3	3	O + C	99	94	Unilateral	33	2011	Med-El Ti100	0
29	F	Genetic	P	18	71	71	36	6	4	4	O	109	110	Bilateral sim.	43	2008	Ci24RE	0
30	F	IUI	P	46	-	-	30	1	1	1	C	111	100	Bilateral sim.	30	2013	Digisonic SP	0
31	F	Unknown	P	36	36	36	10	4	1	1	S	120	120	Bilateral sim.	23	2013	Digisonic SP	0
32	F	Cnx 26	S	12	12	12	1	7	5	4	O	110	112	Bilateral sim.	41	2015	Ci422	0
33	M	Genetic	P	36	NR	NR	NR	4	3	2	O + S	116	120	Unilateral	24	2019	NeuroZTI Evo	0
34	F	IUI	P	36	120	36	10	4	2	3	O	112	111	Bilateral seq.	55	2013	Digisonic Evo	0

HL: Hearing loss; Cnx: connexin; IUI: intra-uterine infection; dg: diagnosis; HA: hearing aid; CI: cochlear implant; CAP: Category of Auditory Performance; SIR: Speech Intelligibility Rating; PTA (dB): pure tone audiometry (decibel); Bilateral seq: sequential; Bilateral sim: simultaneous; WSI: speech intelligibility for disyllabic words in quiet; NR: not-reported. ^¥^: S is for severe and P for profound; ^†^: O is for oral, S for sign, C for code. The grey color is for good performers group.

**Table 2 jcm-11-01874-t002:** Auditory performance in quiet with cochlear implant alone (best or first implanted ear for bilateral simultaneous or sequential implantation, respectively) and communication performance before and 1 year post-implantation in the two groups.

	Poor Performers			Good Performers		
	Preoperative	1 Year Post-CI	*p*-Value	Preoperative	1 Year Post-CI	*p*-Value
Speech intelligibility						
Disyllabic words	4 ± 1.9 (0–30), *n* = 19	14 ± 3.4 (0–40), *n* = 19	0.052 ^¥^	11 ± 4.6 (0–50), *n* = 15	65 ± 4.1 (50–100), *n* = 15	<0.0001 ^¥^
Words in sentences	2 ± 1.6 (0–14), *n* = 9	30 ± 8.6 (0–77), *n* = 12	0.13 ^¥^	20 ± 8.9 (0–92), *n* = 12	75 ± 6.7 (30–100), *n* = 14	0.001 ^¥^
Sentences	0 ± 0 (0–0), *n* = 9	15 ± 6.2 (0–60), *n* = 13	0.25 ^¥^	13 ± 7 (0–80), *n* = 12	64 ± 5.2 (33–100), *n* = 14	0.0007 ^¥^
CAP						
1	2	0	0.015 ^∞^	0	0	0.0003 ^∞^
2	1	1		1	0	
3	4	0		1	0	
4	2	6		1	0	
5	8	4		8	2	
6	0	3		2	3	
7	0	2		1	2	
8	0	1		0	3	
9	0	0		0	4	

Good and poor performers had a speech intelligibility for disyllabic words in quiet ≥ and <50% at 1 year, respectively. CAP: Communication Auditory Performance; CI: cochlear implant. Data are given as mean ± SEM in % (range), and *n* for the intelligibility scores, and *n* for the CAP scores. ^¥^: two-way ANOVA; ^∞^: chi-squared test.

**Table 3 jcm-11-01874-t003:** Evolution in good and poor performers groups before and 1 year after the second implantation in bilateral sequential implantation patients (*n* = 9). Good (*n* = 5) and poor (*n* = 4) performers had a speech intelligibility for disyllabic words in quiet ≥ and <50% at 1 year after the first implantation, respectively (see Table 1).

	Poor Performers (*n* = 4)			Good Performers (*n* = 5)		
	Before the 2nd	1 Year Post-2nd	*p*-Value	Before the 2nd	1 Year Post-2nd	*p*-Value
SI in quiet						
Disyllabic words	33 ± 16.5 (10–70), *n* = 4	50 ± 13.5 (30–90), *n* = 4	0.4 ^¥^	82 ± 11.1 (50–100), *n* = 5	88 ± 6.3 (70–100), *n* = 4	0.9 ^¥^
Words in sentences	48 ± 21.2 (0–94), *n* = 4	66 ± 13.0 (44–89), *n* = 3	0.9 ^¥^	83 ± 9.3 (51–100), *n* = 5	89 ± 10.5 (68–100), *n* = 3	0.7 ^¥^
Sentences	33 ± 17.7 (0–80), *n* = 4	49 ± 17.4 (20–80), *n* = 3	0.8 ^¥^	69 ± 13.6 (33–100), *n* = 5	80 ± 16.6 (47–100), *n* = 3	0.5 ^¥^
SI in noise						
Words in sentences	25 ± 18.4 (0–78), *n* = 4	38 ± 14.4 (11–60), *n* = 3	0.9 ^¥^	55 ± 18.6 (3–91), *n* = 5	78 ± 11.7 (56–100), *n* = 4	0.1 ^¥^
Sentences	15 ± 15.0 (0–60), *n* = 4	20 ± 7.5 (7–33), *n* = 3	0.9 ^¥^	41 ± 17.4 (0–80), *n* = 5	62 ± 20.1 (27–100), *n* = 4	0.07 ^¥^
APHAB	45 ± 4.8 (33–57), *n* = 4	45 ± 4.7 (37–57), *n* = 4	>0.9 ^‡^	48 ± 7.1 (33–67), *n* = 5	39 ± 8.6 (15–57), *n* = 4	0.6 ^‡^
CAP						
5	1	1	0.8 ^∞^	1	0	0.09 ^∞^
6	0	0		1	0	
7	0	1		1	0	
8	2	1		1	2	
9	1	1		1	2	

SI: Speech intelligibility; CAP: Communication Auditory Performance. Data are mean ± SEM (range) in % for the intelligibility scores and the APHAB score, and *n* for the CAP score. ^¥^: two-way ANOVA; ^∞^: chi-squared test; ^‡^: Wilcoxon test.

## Data Availability

The data that support the findings of this study are available on re-quest from the corresponding author. The data are not publicly available due to privacy restriction.

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
