# Peer review of "Candidacy for Cochlear Implantation in Prelingual Profoundly Deaf Adult Patients"

_jcm, 2022, doi:10.3390/jcm11071874_

Round 1

Reviewer 1 Report

This is a significant article for both cochlear implant surgeons and patients alike. We usually resist implanting adult prelingual deaf patients fearing very poor results in terms of any speech production due to poor neural plasticity in adults. These results show encouraging results but require very strict patient selection criteria. Few queries which need some clarity

a) Were these patients self-paid or were covered by government funds? This is very important because cochlear implants are costly and government funds do not cover adults who are prelingual deaf citing poor results. 

b) Did any user become non user in long term especially for the PP group?

c) In page number 7 the graph of poor performer dots are better performers that good performer "squares". Kindly clarify the graph.

Otherwise the encouraging results cited in the study can pave way for further extending the candidacy criteria of cochlear implantation.

Author Response

This is a significant article for both cochlear implant surgeons and patients alike. We usually resist implanting adult prelingual deaf patients fearing very poor results in terms of any speech production due to poor neural plasticity in adults. These results show encouraging results but require very strict patient selection criteria. Few queries which need some clarity

  1. Were these patients self-paid or were covered by government funds? This is very important because cochlear implants are costly and government funds do not cover adults who are prelingual deaf citing poor results. All patients were covered by governments funds. In France, cochlear implant is totally covered by the social security system. This point has been added in the material and methods paragraph, line 71 : “All surgeries and devices were covered by government funds”.
  2. Did any user become non user in long term especially for the PP group? No patient completely abandoned the use of its cochlear implant in the long term. In the PP group, two patients used their device only time to time. I clarified this point line 233-235 : “Among the 34 implanted patients, 32 (94%) were all day long users, and time to time for the 2 others who were in the PP group. No patient has abandoned the device in long term.”
  3. In page number 7 the graph of poor performer dots are better performers that good performer "squares". Kindly clarify the graph. Thank you for this comment, there is indeed a mistake in the graph legend. I changed it for the good one.

Otherwise the encouraging results cited in the study can pave way for further extending the candidacy criteria of cochlear implantation.

Reviewer 2 Report

I do believe this manuscript has its novelty and could attract the readers attentions, but need few more clarification to be made. 

First, this manuscript does not definitively suggest the indication for CI in profound congenital hearing loss adults. After reading the manuscript I cannot determine the candidacy of CI in such cases. I recommend the authors to specify the test tools and outcomes which are favorable. 

Second, please provide the actual hearing thresholds at the diagnosis of congenital hearing loss. Authors mentioned that they excluded progressive hearing losses, but does not provide how they are determined the 'progressive hearing loss'. Since Gjb2 or other genetic cases show progressive  nature this should be clarified. 

Author Response

I do believe this manuscript has its novelty and could attract the readers attentions, but need few more clarification to be made.

First, this manuscript does not definitively suggest the indication for CI in profound congenital hearing loss adults. After reading the manuscript I cannot determine the candidacy of CI in such cases. I recommend the authors to specify the test tools and outcomes which are favorable.

Thank you for this comment. We added a decision algorithm that we propose based on this retrospective study: Figure 4 and line 374-376 : “Based on these results, we propose a decision algorithm for cochlear implantation in adulthood in the case of prelingual deafness (Figure 4).”

Second, please provide the actual hearing thresholds at the diagnosis of congenital hearing loss. Authors mentioned that they excluded progressive hearing losses, but does not provide how they are determined the 'progressive hearing loss'. Since Gjb2 or other genetic cases show progressive  nature this should be clarified.

The patient’s selection was done based on the medical file data. In all cases, we endeavored to recover the initial medical reports, made in early childhood, but these reports, even if they mentioned the profound or severe nature of the deafness, did not necessarily include the data of the audiogram. This is a retrospective study, and this is obviously a bias, but which we all have in consultation when we receive a patient with a very old diagnosis of deafness. A comment has been added in the discussion part, ine 385-386 : “Also, a selection bias is possible since the diagnosis of hearing loss was made many years ago.”

Reviewer 3 Report

The authors present a study on 34 CI patients with prelingual onset profound hearing loss. They provide outcomes, preoperative factors which could predict auditory performance following CI, and feasibilities of bilateral CI. This is an informative paper and provides valuable information. 

Major: I do not think that patients who have profound hearing loss could use oral communication with hearing aids only. How were hearing thresholds with hearing aids in patients? Also, I would like to know why 20 patients requested CI although they could use oral communication for a long time? What was the patient's motivation for CI? The authors need to provide more information.

Minor: In Figure2, do black circle and white square indicate GP and PP, respectively?

Author Response

The authors present a study on 34 CI patients with prelingual onset profound hearing loss. They provide outcomes, preoperative factors which could predict auditory performance following CI, and feasibilities of bilateral CI. This is an informative paper and provides valuable information.

Major: I do not think that patients who have profound hearing loss could use oral communication with hearing aids only. How were hearing thresholds with hearing aids in patients?

Indeed, a profound prelingual hearing loss can not allow the use of oral communication, but in this population, some patients had some benefit with their hearing aid alone before the cochlear implant, as we can see it in the preoperative speech intelligibility for words and sentences (Table 2). Concerning hearing thresholds with hearing aids, this test is not part of the systematic assessment done before cochlear implantation in our center, contrary to speech intelligibility assessment, but it is likely that the prosthetic benefit in pure tone audiometry was sufficient to develop oral communication, in combination with intensive speech therapy in childhood. We added a comment in the discussion part, line 357-359 : “In this study, most of patients used oral communication, alone or in combination with signed language, showing probably some benefit of their hearing aid and a history of intensive speech therapy in childhood’

Also, I would like to know why 20 patients requested CI although they could use oral communication for a long time? What was the patient's motivation for CI? The authors need to provide more information.

Thank you for this question. A psychologist systematically analyzed patients’ motivations before the intervention. The population concerned is a young population, which has grown up and adapted with its deafness, but for which the environment has often evolved: start of higher education or of new work, meeting cochlear implant recipients, starting a family… These evolutions can destabilize the deaf young adult who then have more difficulties, and a clear demand for a cochlear implant, even if he use oral communication. We added a comment in the discussion part, line 371-374 : “Finally, patient’s motivation has to be deeply analyzed before cochlear implantation. In this study, patients came with a request for improved communication, often in the context of a change in their everyday life, such as starting higher education, new work, family, of after meeting cochlear implant recipients”

Minor: In Figure2, do black circle and white square indicate GP and PP, respectively?

Thank you for this comment, there is indeed a mistake in the graph legend. I changed it for the good one.

Round 2

Reviewer 2 Report

Do not have further comment.

Reviewer 3 Report

Thanks for your revision.